# Module-Type Triboelectric Nanogenerators Capable of Harvesting Power from a Variety of Mechanical Energy Sources

**DOI:** 10.3390/mi12091043

**Published:** 2021-08-29

**Authors:** Jaehee Shin, Sungho Ji, Jiyoung Yoon, Jinhyoung Park

**Affiliations:** 1School of Mechatronics Engineering, Korea University of Technology & Education, Cheonan 1600, Chungjeolro, Korea; wogmlchs@koreatech.ac.kr (J.S.); tjdghtmsk@koreatech.ac.kr (S.J.); 2Safety System R&D Group, Korea Institute of Industrial Technology, 320 Techno sunhwan-ro, Yuga-myeon, Daegu 42994, Dalseong-gun, Korea; gonji82@kitech.re.kr

**Keywords:** energy harvesting, triboelectric nanogenerators, vibration energy

## Abstract

In this study, we propose a module-type triboelectric nanogenerator (TENG) capable of harvesting electricity from a variety of mechanical energy sources and generating power from diverse forms that fit the modular structure of the generator. The potential energy and kinetic energy of water are used for the rotational motion of the generator module, and electricity is generated by the contact/separation generation mode between the two triboelectric surfaces inside the rotating TENG. Through the parametric design of the internal friction surface structure and mass ball, we optimized the output of the proposed structure. To magnify the power, experiments were conducted to optimize the electrical output of the series of the TENG units. Consequently, outputs of 250 V and 11 μA were obtained when the angle formed between the floor and the housing was set at 0° while nitrile was set as the positively charged material and the frequency was set at 7 Hz. The electrical signal generated by the module-type TENG can be used as a sensor to recognize the strength and direction of various physical quantities, such as wind and earthquake vibrations.

## 1. Introduction

As the applications of Internet of Things sensors increase, much attention is given to energy harvesting, which is a technology that enables self-generation [1]. Generators that adopt energy harvesting technology for using wasted energy include thermoelectric, piezoelectric, electromagnetic, and triboelectric nanogenerators (TENGs) [2,3,4,5,6]. The triboelectric nanogenerator operates by the movement of electrons from one object to another when two different objects are rubbed against each other. The polarity of the charges on the surface of the contacting triboelectric materials is determined by the triboelectric series, and the surface charges differ according to the position of the external circuit, thus inducing the flow of electrons. Among materials with different properties, negatively charged materials tend to gain electrons, whereas the positively charged materials tend to lose electrons [7,8,9]. Triboelectricity can be used not only for power generation but also for sensors. The application of triboelectric generators as sensors is possible because the amount of electricity generated during friction depends on the degree of external friction [10,11,12,13]. The triboelectric nanogenerator-based sensors are self-generated and will be suitable for various applications, such as motion, vibration, pressure, and tactile sensors. [14]. Additionally, the strength of ocean waves can be monitored by floating a circular triboelectric generator device in the ocean. When multiple devices are combined, electric signals of a wide frequency range and high outputs can be achieved compared with when a single device is used [15,16,17,18,19]. Finally, by using wind power, the direction of the wind can be detected [20,21,22,23] and the amount of rainfall can be measured [24,25,26,27,28]. 

It can be seen from these previous works that researchers have aimed to harvest kinetic energy effectively, but little research has been conducted on a module-type TENG that enables the harvesting of various types of kinetic energy rather than a single type. In this study, we propose a TENG in modular form, which consists of silicone rubber balls and wires embedded inside a cylindrical frame. In the module-type TENG, both the positively and negatively charged materials are placed in the cylindrical housing to form a single structure, unlike in the TENG of previous studies in which negatively and positively charged materials are separated to form two structures. The proposed modular form enables versatility in the application of the TENG because it can be attached to various structures of objects that generate mechanical energy. To demonstrate the capability of the developed module-type TENG in harvesting various types of kinetic energy, an output performance test was conducted in vertical, pendulum, and rotational motions. 

## 2. Materials and Methods

### 2.1. Fabrication Process of the Triboelectric Nanogenerator (TENG) Device and Silicone Rubber Balls

Figure 1a illustrates a schematic of a TENG in a dynamic state. The housing and mold that were used to create the silicone rubber balls of the TENG were created by the fused deposition modeling (FDM) method using a 3D printer (DP203 3D WOX1, Sindoh, Seoul, Korea). Polylactic acid (PLA) was used as the printing filament. To determine the effect of the design and experimental variables, the housing had an outer diameter of 90 mm and a height of 105 mm; an inner diameter of 80 mm was covered with 1 mm of aluminum tape and 1 mm of nitrile. The angle formed by the housing with the floor was set at 0°, 30°, 60°, and 90°. This size was optimized to improve the output performance of the TENG.

As shown in Figure 1b, the interior housing of the TENG was covered with an electrode and a positively charged material in that order. Aluminum tape was used as the electrode. The aluminum, nitrile, and paper were each used as positively charged materials to analyze their effects on output. To create the ball that was used as the negatively charged material, a magnetic bar was used to stir a 1:1 ratio of part A (D.S. NV10) and part B (curing agent) for 5 min. A copper wool electrode of 0.9 g was mixed in the properly stirred silicone rubber; afterward, 17 g was poured into the 3D-printed mold created using the FDM 3D printer and dried for approximately 6 h at room temperature (23 °C). Once created, the composite was inserted into the TENG housing, and a normal single wire (outer diameter of 1.3 mm) was connected to the top plate of the housing to create a pendulum motion, as illustrated in Figure 1a.

### 2.2. TENG Device Assessment

To measure the output performance of the constructed TENG device, we used an oscilloscope (TBS 2072, Tektronix, Oregon, U.S.A.), a voltage probe (P5100A, Tektronix), and a current preamplifier (DLPCA-200, Femto, Berlin, Germany), as shown in Figure 1c. The maximum output was determined by applying resistance from 1 MΩ to 1 GΩ using a variable resistor. Additionally, we conducted an experiment in which we used a digital speed controller (subseries, SPG, Incheon, Korea) to create a frequency of 3–7 Hz for applying a pendulum motion to the TENG. The experiment also enabled us to determine the features that yielded an optimal dynamic behavior for the TENG device.

## 3. Results and Discussion

### 3.1. Preparation and Principle of the Potential Energy–TriboElectric NanoGenerator (P-TENG)

A schematic of the TENG and its dynamic behavior are illustrated in Figure 1, and detailed information is provided in the experimental section. The silicone rubber balls that were made using the mold were attached to the top plate of the housing using a normal single wire and located at approximately 70 mm to create a pendulum motion. As shown in Figure 1b, the housing of the TENG consists of a PLA filament, aluminum tape, and nitrile. For the silicone rubber ball, copper wool was mixed into the silicone rubber interior to improve the output. As the copper wool was 400 µm thin, adding it to the interior of the silicone rubber yields a higher dielectric constant and capacitance than that of the normal silicone rubber. Hence, it is possible to obtain a higher output value because of the greater charge density that is transferred to the electrode [29,30]. Figure 1d,e illustrate the output voltage and current of the TENG. To perform the output tests, the input frequency of the digital speed controller was fixed at 7 Hz, and the resulting output was a voltage of 210 V and a current of 9 μA. The output electricity generation can be described as the charge transfer that occurs between the silicone rubber ball and the Al electrode during physical contact. Figure 2 illustrates the mechanism of current generation via charge transfer when the TENG operates in contact and separation modes. As shown in Figure 2a, charge transfer does not occur in the completely separate initial state or the complete contact state, because electrical neutrality is maintained. However, while the silicone rubber ball and the positively charged electrical materials inside the housing are in the process of making contact or separation, charge transfer occurs to achieve electrical neutrality. During this time, the silicone rubber ball is charged as the negative pole, which has the quality of obtaining electrons easily, whereas the nitrile and aluminum are charged as the positive pole, which has the property of losing electrons easily. Owing to the opposite polarity of the two materials, charge transfer occurs via an external circuit to achieve electrical neutrality, and current is generated. Figure 2b is a modeling image for performing electromagnetic field analysis using the finite element analysis (FEA) technique of COMSOL Multiphysics.

To provide proof of the process described above, the results of an electrical field analysis using the COMSOL program are shown in Figure 2c. The figure illustrates the changes in the electrical field according to the location of the silicone rubber ball. When the space (d) between the silicone rubber ball and the nitrile is 1 mm, a potential difference occurs in the contact part, and a voltage of approximately 200V is produced. This supports the claim that electron movement occurs when contact and separation occur between the silicone rubber ball and the interior of the housing.

### 3.2. Electrical Performance of the P-TENG

Figure 3 shows the process of optimizing the electrical output characteristics when the TENG operates in the vertical contact separation mode. To obtain optimal data for the conditions under which the TENG device operates, experiments were performed using different frequencies. The experiments analyzed the voltage and current data that were produced as the input frequency of a digital speed controller were changed from 3 to 7 Hz. At a frequency of 3 Hz, a voltage (V) of 10 V and a current (I) of 3 μA were obtained. It was found that the voltage and current both increased linearly with frequency. Based on this, it is observed that the contact area increases, and the output voltage and current increase because of the increase in the force acting on the silicone rubber ball and the motion energy as the frequency increases. At 4 Hz, there was a V of 4 V and an I of 4.5 μA, and at 5 Hz, there was a V of 150 V and an I of 7 μA. At 7 Hz, the V and I were 250 V and 11 μA, respectively. The experimental results show that the output values rose sharply, starting at 5Hz. This occurred because as the frequency increased, the amplitude of the pendulum motion of the silicone rubber ball increased, and it was possible for the top and bottom surfaces of the TENG housing interior to make contact effectively. Owing to this effect, the output at 7 Hz improved by more than 200 V compared with that at 3 Hz, and the I increased by more than 8 μA. This was proven by the governing equation for a contact- and separation-mode TENG, as shown below.
(1)$$E=\frac{Q}{S\varepsilon_{0}\varepsilon_{r}},$$
(2)$$V=-\frac{\sigma }{\varepsilon_{0}}\;\left(\frac{d}{\varepsilon_{r}}+x\left(t\right)\right)+\frac{\sigma x\left(t\right)}{\varepsilon_{0}},$$
(3)$$I=-\frac{S\sigma \frac{d}{\varepsilon_{r}}v\left(t\right)}{{\left(\frac{d}{\varepsilon_{r}}+x\left(t\right)\right)}^{2}},$$
where Q is the value of the transferred charges between the two electrodes, S is the dielectric area size, $\varepsilon_{0}$ is the vacuum permittivity,$\;\varepsilon_{r}$ is the relative permittivity, σ is the friction charge surface density, x(t) is the distance between two contact surfaces, t is the time, and d is the effective dielectric thickness. According to Equations (1)–(3), when the frequency increases, the pendulum motion amplitude of the silicone rubber ball increases, and the contact area with the surface of the housing interior increases, which increases the wattage and current. Figure 4 shows the normal form of the silicone rubber ball compared with those that had dimple and bump patterns newly applied to them in order to effectively increase their contact area. Figure 4a shows a comparison of the four shapes of the silicone rubber balls. They are only-D.S. ball, D.S.+Cu wool ball, dimple ball, and bump ball, in that order. Figure 4b shows a graph illustrating the analysis of the changes in the voltage and current values according to the differences in the ball shape. As shown in Equations (1)–(3), the electrical output performance of the TENG is significantly affected by the contact area. Figure 4 shows the normally shaped silicone rubber ball compared with silicone rubber balls that have newly applied dimple and bump-shaped patterns to effectively increase their contact area. 

It is observed that the D.S.+Cu wool ball, dimple ball, and bump ball did not show large changes in voltage and current, but the only-D.S. ball showed markedly lower voltage and current values. The reason for this is because the only-D.S. ball was the only one of the four shapes that operated in the single-electrode mode because it did not contain Cu wool that could act as an electrode. It is well known that the single-electrode mode has lower electrical output performance than those of the contact and separation modes. Therefore, it is observed that the only-D.S. ball obtained markedly lower output performance compared with that of the other ball shapes. Cu wool was added to achieve high capacitance and specific inductive capacity in order to receive a high charge density from the electrode so that a high output value could be produced. In the dimple model, a dimple pattern was applied to the surface to increase the contact area, but when the model was used in practice, it was found that the generated contact area was slightly reduced, which had an adverse effect on the output value. It is expected that the contact area will increase and the output performance will improve if an external force makes an extreme change to the shape of the ball. However, within the frequency range of the experiment, it is expected that there will be no major effect, because the outer shape of the silicone rubber ball was not greatly changed. In the case of the bump shape, wire shapes with a diameter of 0.6 mm and a height of 1 mm performed the role of improving the contact area when making contact with the surface of the housing interior. As shown in Figure 4b, the highest output voltage and current were both obtained by the ball with bump shapes, which resulted in a V of 210 V and an I of 9 μA.

Figure 5 illustrates a comparison and analysis of the changes in the output according to the size and quantity of dynamically behaving rubber balls in the TENG. First, to select the optimal size for the silicone rubber balls, the mold was used to prepare 10 pi, 20 pi, and 30 pi balls, as shown in Figure 5a. Balls of sizes less than 10 pi have an insufficient mass to facilitate smooth operation of contact/separation; thus, 20 pi and 30 pi balls were also prepared based on the 10 pi ball. In addition, as the inner diameter of the housing is 80 mm, when a ball of 30 pi or larger is used, although the contact area increases and the output will increase, the contact/separation process will not be performed smoothly. Their masses were 0.73 g, 5.44 g, and 15.8 g, respectively, and the masses of the copper wool within the silicone rubber balls were 0.2 g, 0.4 g, and 0.9 g, respectively. As shown in Figure 5b,c, it was found that the voltage and current values increased as the sizes of the silicone rubber balls increased. This can be explained by Equations (1)–(3). As the diameter increased, the friction charge surface density (σ) value increased along with the capacitance (C) value. Consequently, the 30 pi silicon rubber ball created more charge transfers than those of the 10 pi ball, and it was able to produce a large output value. From the actual measurement results, the 10 pi ball had a V of approximately 30 V and an I of 2 μA, whereas the 30 pi ball had a V of 200 V and an I of 10 μA. Figure 5d shows a schematic of silicone rubber balls connected in quantities of 1 to 4. The experimental conditions were set up so that the balls were fixed at 70 mm on the TENG housing interior and a frequency of 7 Hz was applied, as in the experiments with a single silicone rubber ball. As the number of silicone rubber balls increased, there was no substantial difference in V_max_ or I_max_. However, by observing the voltage and current waveforms, it can be seen that the density improved, and this is shown in Figure 5e. By observing the waveforms for a single silicone rubber ball, it can be seen that a single output was produced during the pendulum motion. Similarly, it can be seen that in the case of two balls, two outputs were produced for a single pendulum motion, while three and four balls produced three and four outputs, respectively. This helps to increase the V_rms_ and I_rms_ values because the density in the waveform is increased. From the data values, one ball had a V_rms_ of 33.71 V and an I_rms_ of 0.904 μA, two balls had a V_rms_ of 39.954 V and an I_rms_ of 0.948 μA, three balls had a V_rms_ of 41,869 V and an I_rms_ of 1.078 μA, and four balls had a V_rms_ of 45.825 V and an I_rms_ of 1.27 μA. Thus, it was found that the V_rms_ and I_rms_ values increased as the number of balls increased.

### 3.3. Properties of Location and Material

As shown in Figure 6, the output at each angle of the P-TENG was analyzed. Experiments were performed to examine changes in the location of the TENG due to changes in the environment as well as changes in the state of the TENG due to the applied external force. As shown in Figure 6a, the experiments were performed on frames built with angles of 0°, 30°, 60°, and 90°; the location of the TENG was fixed at 70 mm inside the housing and the experiment was performed at a frequency of 7 Hz.

In Figure 6b,c, the highest values (200 V and 9 μA) were obtained at 0°. Conversely, the lowest output values (40 V, 1.5 μA) were obtained at 90°. The output values were the lowest at 90° because at 0° it was possible to make effective contact and separation on both sides of the housing during the pendulum motion, whereas at 90°, a lower output value was obtained because of a smaller contact area in comparison with that for 0°. Next, changes were made to the material of the positively charged materials inside the cylinder. A comparative analysis was performed on the effective positively charged materials following the triboelectric series, which are commonly encountered in daily life. Positively charged materials were created using Al, Al + nitrile, and Al + paper, as shown in Figure 6d. The most basic way to improve the output performance of a TENG is to increase the surface potential difference. Depending on the choice of the positively charged materials used, the difference in surface potential can be increased. In Figure 6e,f, when comparing the three materials, the output of Al + nitrile was similar to that of Al and Al + paper, but it was confirmed that the RMS value was the largest, and thus Al + nitrile was selected as the positively charged material of P-TENG.

### 3.4. Application

As shown in Figure 7, the output values found in the experiments were used to light LEDs that can operate at low power. For experiments presented in Figure 7a,d, the housing applied with nitrile and the 30 pi bump ball were used at a frequency controlled at 7 Hz with a digital speed controller. For the experiment shown in Figure 7e, the P-TENG was fixed at the end of the cantilever structure, and pendulum motion was performed in the left–right direction. To check the electrical power produced by the P-TENG, measurements were taken at a variable resistance of 1 MΩ to 1 GΩ, as shown in Figure 7a. As the resistance increased, the voltage tended to increase and the current decreased. The results of calculating the output power values for each external resistance value showed that the maximum power values were 460 µW at 100 MΩ. As shown in Figure 7b, a bridge rectifier was used to assess the charging characteristics of the equipment, and the AC signal was converted into a DC signal. The capacitors used in the charging tests had the following capacitances: 0.82, 1, 1.5, 2.2, 3.3, 4.7, and 10 nF. Figure 7c shows a graph of the charging curve resulting from the use of a variety of capacitors under the same input conditions. The time required to charge the 10 nF capacitor to 14 V was 2.5 s. In Figure 7c, a 10 nF capacitor and LEDs were connected to the rectifier circuit to conduct a charge/discharge test. The AC signals from the P-TENG was converted into DC signals with a rectifier circuit, and the 10 nF capacitor was charged to light the LEDs. When the LEDs were turned on, the charge stored in the capacitor was discharged.

Figure 7d is a graph of the charge/discharge curve. To verify the performance of the P-TENG as an actual power source, a cantilever and an LED electric circuit were set up, and Figure 7e shows the 30 LED lights. The proposed P-TENG has a module-type structure for harvesting various kinetic energies. As a P-TENG generates outputs using a mechanism of contact and separation modes, the experiment was conducted with a digital speed controller of a vertical motion structure that allows the most effective operation of contact/separation of the silicone rubber ball. A frequency of 7 Hz was applied for the experiment, and 30 LEDs were turned on with a peak output of 250 V and 11 µA. These details are illustrated in Figure 7d. The proposed P-TENG is capable of harvesting various types of kinetic energy by being attached to a structure that performs pendulum motion or rotational motion, in addition to the vertical motion. In Figure 7e, the P-TENG was fixed at the end of the cantilever structure for pendulum motion, and the peak output of 130 V was obtained. In Figure 7f, the P-TENG was attached to a rotating body for harvesting rotational kinetic energy. The peak output of 80 V was obtained from the experiment. In this way, it was demonstrated that it is possible to obtain electrical output performance after attaching the proposed modular type P-TENG to various types of kinetic energy sources.

## 4. Conclusions

In this study, we proposed and implemented a P-TENG, which is a pendulum-motion-based TENG that effectively collects the potential energy of a ball. While the negatively charged and positively charged materials are separated and an external device performs the contact/separation of the charged materials in the conventional setup, in the proposed module-type P-TENG, a silicone rubber ball (the negatively charged material) and a nitrile rubber (the positively charged material) are both placed inside the housing. This setup enables contact/separation without the need of an external device. As no external device is required, the P-TENG can be freely attached to various places to convert mechanical energy sources to electrical energy for power supply. To improve the generation performance, a copper wool electrode was inserted into the silicone rubber ball, thereby increasing the capacitance and achieving a higher output. Additionally, the dynamic behavior of the P-TENG was analyzed, and the design variables that affected the output were identified. The shape, size, and quantity of the silicone rubber ball significantly affected the contact area with the interior surface of the housing during dynamic behavior. To effectively improve the contact area, four new silicone rubber ball shapes (only-D.S. ball, D.S.+Cu wool ball, dimple ball, and bump ball) were proposed, and it was found that the bump ball provided the highest generation performance. Additionally, it was found that the generation performance increased as the size and quantity of balls increased, and this is believed to be due to the increase in friction surface charge density and capacitance. Finally, the P-TENG device was used as a power source to light 30 LEDs and power commercial electronic products, which proved that the kinetic energy created by the potential energy of the ball can be effectively harvested as electrical energy. Varying output values can be produced when different angles are assigned to the TENG, and its shape status changes for each angle. By using these output values, it is possible to create a monitoring sensor that can detect the functional status and deformation states without directly checking the status. Therefore, the results of this study may be used for wind volume and wind speed measurement sensors in the future. This is expected to have two effects because a TENG can be used in a variety of situations as a module type.

## Figures and Tables

**Figure 1 micromachines-12-01043-f001:**
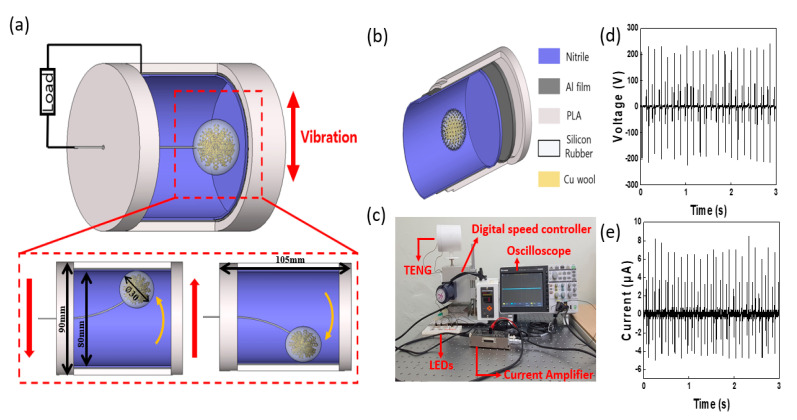
Schematic and experimental configuration of a triboelectric nanogenerator (TENG). (**a**) Schematic and movement of the TENG in a dynamic state, (**b**) material composition of the TENG, (**c**) photograph of the configuration of the experimental equipment, (**d**) plot of the output voltage, and (**e**) plot of the output current of the TENG. PLA: polylactic acid.

**Figure 2 micromachines-12-01043-f002:**
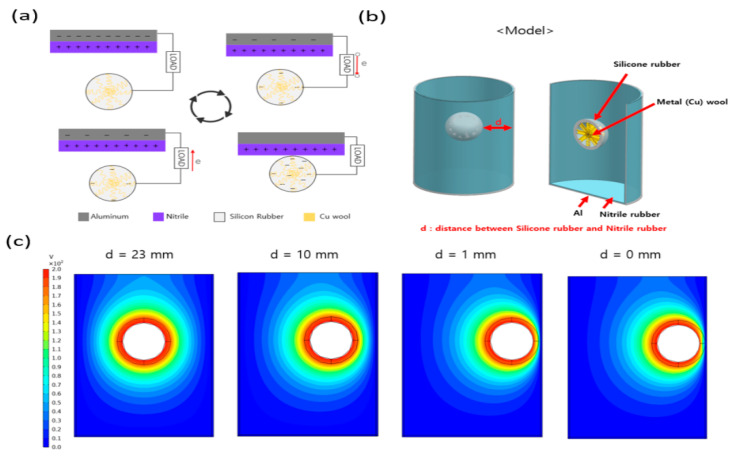
Contact- and separation-mode mechanism and COMSOL simulation of the P-TENG cylinder and Dragon Skin (D.S.) ball: (**a**) schematic of the contact- and separation-mode mechanism, (**b**) diagram showing modeling for the simulation, and (**c**) gradient image depicting COMSOL simulation results.

**Figure 3 micromachines-12-01043-f003:**
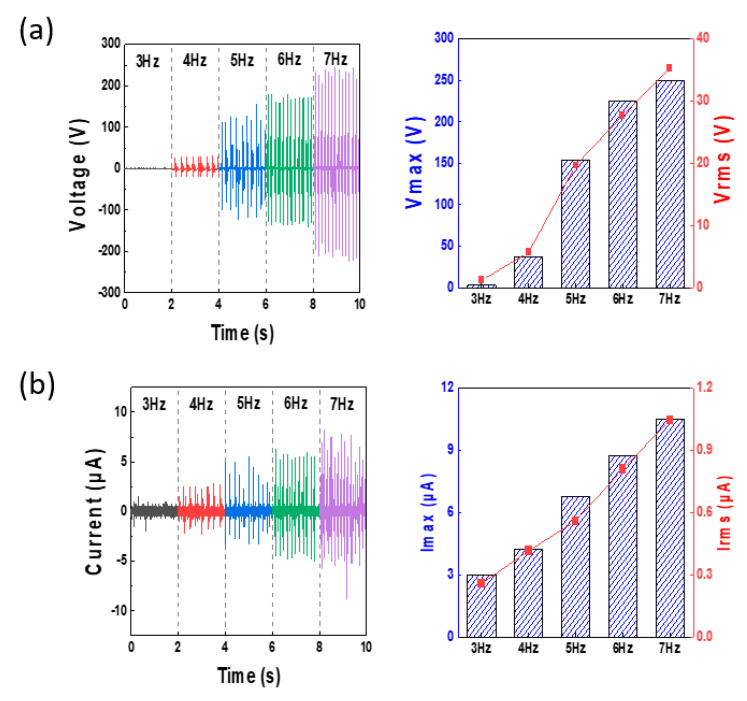
Plots depicting changes in output according to changes in frequency: (**a**,**b**) voltage and current occurring from 3 to 7 Hz.

**Figure 4 micromachines-12-01043-f004:**
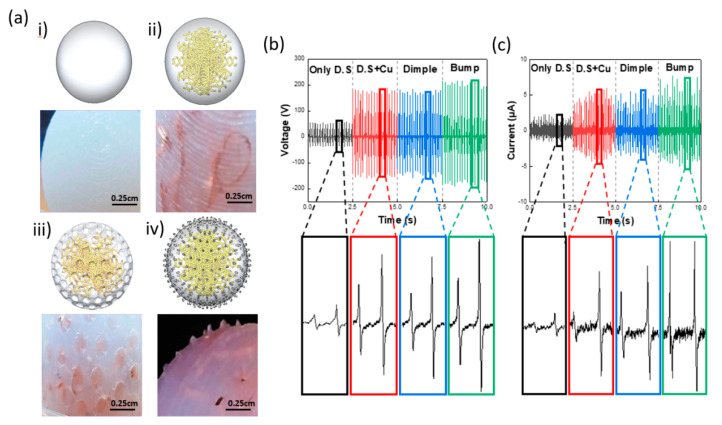
Changes in the output values according to the shape of the silicone rubber ball. (**a**) Pictorial representation and photographs of silicone rubber ball shapes: (i) the modeled shape and a 4× magnification photograph of the silicone rubber surface of the ball made only of D.S., (ii) the modeled shape and a 4× magnification photograph of the silicone rubber surface of the ball made by mixing copper wool into the silicone rubber ball interior, (iii) the modeled shape and a 4× magnification photograph of the silicone rubber surface of the ball made by adding dimple shapes to the silicone rubber ball surface to change the contact area, and (iv) the modeled shape and a 4× magnification photograph of the silicone rubber surface of the ball made by adding bump-shaped wires to the surface of the silicon rubber ball to change the contact area. (**b**,**c**) Plots showing the voltage and current according to differences in the ball shape.

**Figure 5 micromachines-12-01043-f005:**
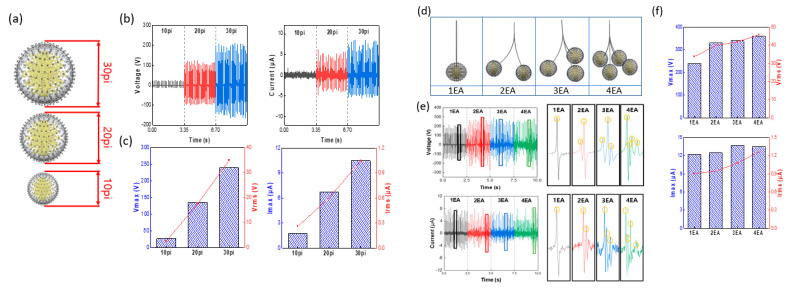
(**a**) Pictorial representation of various sizes of silicone rubber balls; (**b**,**c**) plots depicting *V*_max_, *V*_rms_, *I*_max_, and *I*_rms_ according to size; (**d**) schematic showing a comparison of quantities of silicone rubber balls; and (**e**,**f**) plots of the voltage and current output waveform, and *V*_max_, *V*_rms_, *I*_max_, and *I*_rms_ according to quantity.

**Figure 6 micromachines-12-01043-f006:**
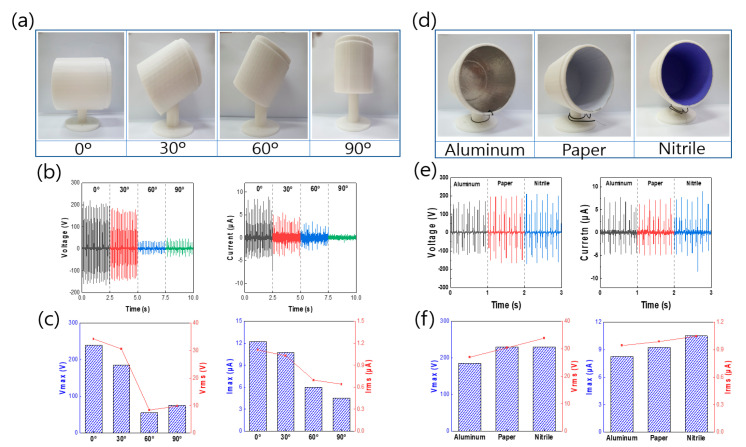
**(a**) Photographs showing the angles of the TENG cylinders; (**b**,**c**) plots of V_max_, V_rms_, I_max_, and I_rms_ according to the cylinder angles; (**d**) photographs of the types of positively charged electrical materials on the TENG interior surfaces; and (**e**,**f**) plots of *V*_max_, *V*_rms_, *I*_max_, and *I*_rms_ values according to the type of positively charged electrical materials.

**Figure 7 micromachines-12-01043-f007:**
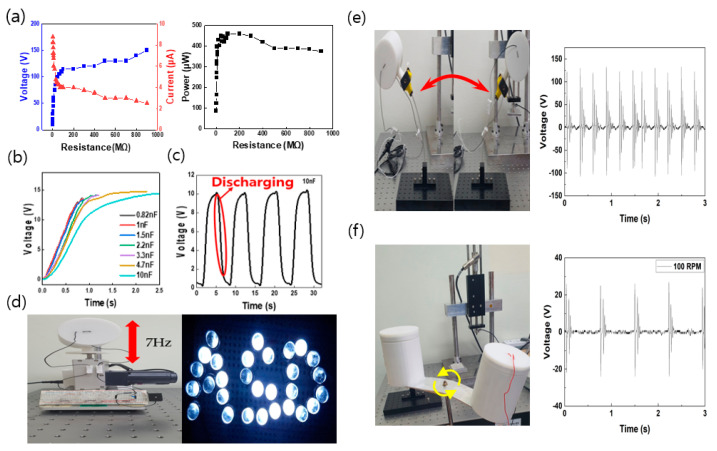
Application: (**a**) Plots depicting the influence of load resistance on the P-TENG, (**b**) plot depicting the use of capacitors with different capacitance values (0.82, 1, 1.5, 2.2, 3.3, 4.7, and 10 nF), (**c**) plot of the curve of charge/discharge when two LEDs were connected, (**d**) image of the “ASD” pattern LED by the P-TENG (photograph of the digital speed controller and electrical circuit configuration), (**e**) pendulum kinetic energy, and (**f**) rotational kinetic energy.

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
