# Peer review of "Module-Type Triboelectric Nanogenerators Capable of Harvesting Power from a Variety of Mechanical Energy Sources"

_micromachines, 2021, doi:10.3390/mi12091043_

Round 1

Reviewer 1 Report

This manuscript presents an interesting module-type triboelectric nanogenerator (TENG), which can harvest power from several mechanical energy sources. The electricity is generated by the contact/separation mode between two triboelectric surfaces inside the TENG. The module-type TENG has potential application as sensor to identify the strength and direction of physical parameters, including wind or earthquake vibrations. However, this manuscript must be improved considering the following comments:

1.-English style and grammar of manuscript should be checked.

2.-Abstract section should include an introduction sentence about the research problem. In addition, this section should include the materials and main results of TENG.

3.-Introduction section must add more  discussion of the main advantages and limitations of the recent works related with triboelectric nanogenerators. Authors must include the advantages and limitations of their module-type TENG in comparison with other TENG reported in the literature.

4.-Section of materials and methods. This section should include a schematic view of figure with more detail information about the contact/separation generation mode considering the different layers of TENG. The contain of Figure 1 should be improved.

Section 2 is short, and it should add more detail information about the different dimensions of components, materials, connections, and experimental setup of TENG. Authors should consider a table with the dimensions of the  components and materials of TENG. In addition, authors must mention more information of the design and fabrication process of the components of their module-type TENG, including the silicon rubber ball. This information should be included in section 2.

5.-Authors must add more data about the FEA models using COMSOL. For instance, element types, boundary conditions, and mesh type.

6.-Which was the criterion to select a frequency range from 3 to 7 Hz?

7.- Which was the criterion to select the materials aluminum, paper and nitrile of TENG?

8.- Which was the criterion to select the different sizes of silicone rubber balls?

9.-Figure 3(a), Figure 4(b-c), Figure 5(b,e), and Figure 6(b,e) should include better views of the results of voltage and current.

10.-Authors must add more discussion about the results of Figure 5, 6 and 7.

11.-Quality of Figure 7 must be improved.

12.-Authors must add more information about the experimental setup used to obtain the results of Figure 7.

13.-Conclusion section must be improved considering the modifications of the revised manuscript.

Author Response

I have attached the answer sheet and the revised thesis.

Reviewer 2 Report

The manuscript titled "Module-Type Triboelectric Nanogenerators Capable of Harvesting Power from a Variety of Mechanical Energy Sources," investigates a TENG method of harvesting energy using a ball on a beam method. Overall TENG is of interest, but the paper needs significant changes before it is ready for acceptance.

1) i do not see the micro-aspect of this so why submit to micromachines? if it was macro  scale and a method of making it into a microdevice is explained i could see. But I don't see how you would make this into a micro-scale device.

2) I'm not sure what the novelty is in this paper the metallic balls are a little, but the overall structure of the energy harvester isn't that novel.

3) Introduction needs major work, the introduction went over TENG and how they are used but did not talk a lot about what other TENGs have done and how this one will be better. In the introduction i didn't see any mention to how the device in the manuscript is better or what the problem was and how this method will solve this.  So overall the introduction didn't introduce the paper after reading it i still didn't know what the authors were presenting and why.

The use of simulation and experimentation data is good.  And Figure 4 demonstrating effects of various materials is good.

3) Dragon skin should be explained what this is, I'm assuming this is the silicone elastomer from Smooth on that is called dragon skin, which I'm familiar with but others might not know what dragon skin is.

4) should give more detail on how the Cu was embedded in the dragon skin

Author Response

(The authors gave the same response as above.)

Round 2

Reviewer 1 Report

This version of manuscript has been improved considering the reviewer's comments. This manuscript is suitable for publication in Micromachines.

Reviewer 2 Report

I have one minor comment. In Figure 1(b)  the legends says silicon rubber, but it should say silicone rubber.